# Sex Differences in Behavior and Learning Abilities in Adult Rats

**DOI:** 10.3390/life13020547

**Published:** 2023-02-15

**Authors:** Maria Pupikina, Evgenia Sitnikova

**Affiliations:** Institute of the Higher Nervous Activity and Neurophysiology of Russian Academy of Sciences, Butlerova Str., 5A, 117485 Moscow, Russia

**Keywords:** sucrose preference, von Frey test, free-choice paradigm, avoidance learning, fear conditioning, ultrasonic vocalization, aversive calls

## Abstract

Laboratory rats have excellent learning abilities and are often used in cognitive neuroscience research. The majority of rat studies are conducted on males, whereas females are usually overlooked. Here, we examined sex differences in behavior and tactile sensitivity in littermates during adulthood (5.8–7.6 months of age). We used a battery of behavioral tests, including the 2% sucrose preference test (positive motivation), a free-choice paradigm (T-maze, neutral situation), and associative fear-avoidance learning (negative motivation, aversive situation). Tactile perception was examined using the von Frey test (aversive situation). In two aversive situations (von Frey test and avoidance learning), females were examined during the diestrus stage of the estrous cycle, and ultrasonic vocalization was recorded in both sexes. It was found that (1) females, but not males, lost their body weight on the first day of the sucrose preference test, suggesting sex differences in their reaction to environmental novelty or in metabolic homeostasis; (2) the tactile threshold in females was lower than in males, and females less frequently emitted aversive ultrasonic calls; (3) in the avoidance learning task, around 26% of males (but no females) were not able to learn and experienced frizzing. Overall, the performance of associative fear-avoidance in males was worse than in females. In general, females demonstrated higher abilities of associative learning and less persistently emitted aversive ultrasonic calls.

## 1. Introduction

The majority of rat studies are performed on males, whereas females are usually overlooked. Recent (2016–2017) analytical research [1] indicated that only 4–16% of all publications on rats included both sexes, and 0–5% were female-only, in contrast to the 74–94% of male-only studies. The prevalence of male-only rat studies in the cognitive domain might cause misleading simplification. In order to disclose sex differences in neurocognitive aspects, here we used a battery of behavioral tests in littermates (males and females) during adulthood (5–8 months of age). First, a positively motivated task (sucrose preference test), then a neutral task (a free left/right choice task), and finally two aversive tasks (von Frey test and active avoidance task) were performed.

Rodents like sweet taste. As early as in 1967, Elliott Valenstein and coauthors demonstrated that female rats of Charles River CD and Holtzman strains consumed larger quantities of a mildly sweet 3% glucose solution and a 0.25% saccharin solution (judged to be very sweet by humans) [2,3]. Since that time, the sucrose preference test has been widely used in rodents in many variations [4,5,6]. Recently, Tapia et al. in 2019 [5] found that female rats worked harder in order to obtain a sucrose pellet and consumed more sucrose pellets than males. Grimm et al. in 2020 [6] showed that female Long-Evans rats consumed more sucrose than males and showed no sex difference in saccharin preference. Furthermore, Laura Buczek et al. in 2020 [7] demonstrated in Sprague Dawley rats that females showed an increased susceptibility to overeat palatable food in comparison to males. In the above cited papers, different behavioral paradigms were used to test sweet preference in rats. At the same time, sucrose preference tests are very common for assessing anhedonia in rodents [5,8,9,10], especially on the preclinical stage of the development of antidepressant drugs, in which depressive-like behavior in rats is associated with low sucrose intake [9,11,12]. Now, if females have a stronger sucrose preference than males, they are less “depressive” than males; thereby, the results of sucrose preference tests in females should be interpreted with caution. In the current study, we evaluated a leading hypothesis that females have a stronger sucrose preference than males based on a standard two-bottle choice paradigm (2% sucrose vs. tap water) for two days.

Neurocognitive functions in females and males might differ as a result of the neuroactive action of gonadal hormones [13,14,15,16]. Testosterone and its metabolites are known to have cognitive-enhancing effects in rodents [17,18,19]. On the other hand, the effect of gonadal hormones on behavior seems to be overestimated [1]; adult female rats behaved in a similar manner to males in the majority of tests (see references in [20]). Ovarian hormones also affect the behavioral indices of anxiety [14,21,22], and this has to be taken into account in the pain-related fear conditioning of females. For example, female rats in the behavioral estrus (determined based on the specific behavioral reactions during contact with a sexually experienced male) performed better in the inhibitory avoidance task than females in diestrus [22]. Additionally, punishment-avoidance learning in Long–Evans female rats was faster than in males [23]. Likewise, in the learned helplessness paradigm in Sprague–Dawley rats, «*females learn to escape the shock much sooner than do males, even without any previous exposure to uncontrollable stress*» [24]. Recent studies have described different behavioral strategies in females and males in fear-related associative tasks [23,24,25,26]. For instance, Tara Chowdhury et al. noted: «*about half of females successfully avoided the shock (“avoiders”), the other half consistently waited for shock to begin before immediately performing the response to turn it off (“escapers”)... Whereas male subjects show a near-uniform elevation in freezing to an aversive conditioned stimulus*». In the current study, we evaluated learning strategies and learning skills in males and females by training rats for classic active avoidance in parallel with the analysis of aversive 22 kHz vocalization.

Rats emit ultrasonic vocalizations in a variety of social situations [27,28,29,30]. Ultrasonic calls in rats could be roughly divided into two classes: short 50-kHz calls in appetitive or “friendly” (i.e., nonaggressive) behavioral situations and long-lasting 22-kHz alarm calls in aversive and potentially dangerous situations [27,29,31,32]. Little is known about sex differences in ultrasound aversive vocalization. Only one study compared ultrasonic vocalization during isolation restraint stress in males and females; in this aversive situation, females in estrus and proestrus emitted fewer 22 kHz aversive calls than males [33]. It is remarkable that the level of aversive 22 kHz vocalization in females in diestrus was the same as that in males. The estrous cycle in female rats typically lasts 4–5 days and includes four stages: proestrus, estrus, metestrus, and diestrus. These stages differ in the levels of estradiol and progesterone secreted by the ovarian follicles. Diestrus is the longest phase (lasting on average 48–72 h) characterized by low levels of estrogen and progesterone [14]. All things considered, we used females in diestrus for aversive tests (von Frey test for sensitivity and fear-conditioned learning).

It is well known that female subjects have a higher tactile sensitivity and are more sensitive to pain than male subjects [34,35,36,37], «*though sex differences in emotional-affective and cognitive responses to pain may not be as clear*» [36]. Sex differences in pain-related fear conditioning paradigms might link to emotional self-regulation [38,39]. In this context, we compared tactile sensitivity and associative fear-avoidance learning in females and males in parallel with recorded ultrasounds. The analysis of ultrasonic vocalization was performed here in order to characterize the emotional state of females and males in aversive situations.

Our study was conducted on a non-epileptic rat substrain derived from Wistar Albino Glaxo Rats from Rijswijk (Non-Epileptic WAG/Rij abbreviated as “NEW”). As the mother strain of NEW rats, WAG/Rij rats are a well-accepted rat model of the absence epilepsy, showing spontaneous spike-wave discharges in their electroencephalogram (EEG) [40,41]. The selection work and breeding of the NEW substrain has been carried out in our institution (Institute of Higher Nervous Activity and Neurophysiology of RAS, Moscow, Russia) since the second decade of the 21st century, when we selected female and male WAG/Rij rats without seizures during the entire life span [42]. NEW rats were selected as a non-epileptic control for WAG/Rij rats and had similar genetic backgrounds. Considering the common genetic background, the current findings in the NEW rats could be applicable to the other Wistar Albino rat strains.

## 2. Materials and Methods

The current study was conducted on the “NEW” rat substrain. The rats were bred at the Institute of Higher Nervous Activity and Neurophysiology of RAS, Moscow. The experiments were carried out in accordance with EU Directive 2010/63/EU for animal experiments and were approved by the animal ethics committee of our Institute (protocol No. 4 approved on 26 October 2021 and additional protocol No. 4 approved on 13 December 2022). Rats were kept in environmentally controlled conditions with a 12:12 h light:dark cycle (light on at 08.00) with constant airing. Rats were housed in same-sex groups (3–4 subjects per cage). Food and water were provided ad libitum.

In total, 50 rats were used (19 females and 31 males). Group 1 consisted of 35 rats (11 females and 24 males), which were siblings from 5 litters born in March–April 2022 (from 5 to 9 pups per litter). Dams were removed after 25–27 postnatal days, and rats were housed in small groups (3–4 same sex rats per cage). The sucrose preference test, T-maze test, and von Frey test were conducted on Group 1 (Figure 1e).

Active avoidance tests were conducted on Group 1 (9 females and 24 males) and on the parent group of 15 rats (Group 2, 8 females and 7 males, Figure 1e). After the end of behavioral experiments, rats from Group 2 were implanted with epidural electrodes for the electroencephalographic recording of free behavior. These rats showed almost no spike-wave abnormalities; therefore, we confirmed that they were non-epileptic (see Appendix A).

### 2.1. Estral Cycle

The phase of the estrous cycle in females was determined in wet smears immediately after collection (direct cytology, unstained slides) with a Nikon microscope [20,43,44]. Vaginal swabs were obtained using sterile saline and examined on a slide in a drop of saline. Microscopic examination was performed with a 10× objective to determine the relationship between cell types and a 40× objective to recognize cell types. 

The swabs contained four types of cells: 

1. Leukocytes (neutrophils or polymorphonuclear cells), which are very small round cells. 

2. Small nucleated epithelial cells, which are small non-keratinizing cells of a round or oval shape. 

3. Large nucleated epithelial cells of a round or polygonal shape with serrated or angular edges.

4. Non-nuclear keratinized epithelial cells or needle-like cells.

Diestrus (the longest phase) lasted on average 48–72 h and was characterized by the predominance of leukocytes. Leukocytes were absent in the stage of proestrus and estrus. Some leukocytes and epithelial cells were detected during metestrus (6–8 h).

### 2.2. Behavioral Tests

A battery of behavioral tests were performed in a sequence from low to high stressfulness:

1. Sucrose preference test (average age: 5.8 months, Figure 1a).

2. A free choice task (T-maze) was used to characterize the preferences of left/right sides in a free choice condition (average age: 6.9 months, Figure 1b).

3. Quantitative sensory testing (von Frei test) was used to measure the sensitivity of hindlimbs to mechanical stimuli (average age: 7.0 months, Figure 1c).

4. Active avoidance test was used to test learning abilities and behavioral strategies in an artificial adverse environment (average age: 7.6 months, Figure 1d). 

#### 2.2.1. Sucrose Preference Test 

The sucrose preference test is one of the most common tests used to assess anhedonia in rodents [12]. During this test, rats were kept in home cages (2 or 3 rats per cage) and were not deprived of food or drink before and during the test. We used a two-bottle choice paradigm, in which rats were presented with two pre-weighed drinking bottles for 24 h. One bottle contained a 2% sucrose solution, and another contained drinking water (Figure 1a). During the 1st day, drinking water was at the usual place (left corner), and the 2% sucrose solution was on the opposite (right) side. On the next day, bottles were weighed and replaced. Rats were weighed before the test, at the end of the 1st day, and at the end of the 2nd day. Two-day dynamics of body weight were examined. 

The sucrose preference was calculated according to the formula: sucrose preference = (sucrose solution consumed)/(sucrose solution consumed + plain water consumed) × 100%.

The volume of consumed 2% sucrose solution was computed, accounting for body weight. Inasmuch as rats were kept 2 or 3 rats per cage, the total body weight was computed and used for the normalization: sucrose consumed = (sucrose solution consumed)/(total body weight).

#### 2.2.2. A Free Left-Right Choice Task (T-Maze)

This simple test was performed in a plastic T-maze consisting of the start arm and two goal arms: left/right (Figure 1b). This test is based on the willingness of rats to explore a new environment, i.e., visit a new arm of the maze or a familiar arm. Here, we accessed exploratory activity and the spontaneous alternation between left and right sides.

Adaptation: Rats were transferred to the experimental room and kept in cages for habituation (minimum 10 min). 

Testing: Each individual rat was placed in the start arm of the T-maze facing the corridor. Upon leaving the start arm, a rat entered either the left or the right goal arm. As soon as it fully entered the goal arm (full body), it was removed from the T-maze and placed back in the transfer cage for 5 s. The test was repeated 5 times. If a rat stayed in the start arm for 60 s, it was moved from the T-maze to the transfer cage for 5 s, and this trial was classified as “passive”. In order to characterize the behavioral response in a free choice condition, we scored the number of “passive” trials and the degree of lateralization (the ratio between left and right choices). 

Analysis: The passive reaction was noted when a rat stayed in the starting arm for 1 min (zero). Results of this test represented a sequence of choices, e.g., RLRRR, RLL0R, LL000. For computational analysis, R was scored as 1; L was scored as −1. The total score or the laterality index was computed as a sum of scores in 5 trials. Passive behavioral lateralization was noted in rats with three or more zeros, and the laterality index was [−1; +1]. In the rest rats, the laterality index varied from −3 (left preference) to +3 (right preference). Left preference was assigned when the laterality index was [−3; −2] or more than two “L” choices. Right preferences were assigned when the laterality index was [+3; +2] or more than two “R” choices.

#### 2.2.3. Test (Von Frey) 

The von Frey test is the one of most commonly used behavioral tests of mechanosensation in rodents. It includes the application of a punctate stimulus to the plantar surface of the hind paw. It is noteworthy that the von Frey threshold of mechanical sensitivity is a measure of an animal’s reaction to a mechanical irritation (aversive), opposing to obvious pain caused, for example, by the Rundell-Sellito apparatus [45,46]. 

Here, we used the Ugo Basile Von Frey Hairs (Semmes-Weinstein set of 20 monofilaments, Aesthesio^®^). Microfilaments were nylon-made and labeled by the “target force” in g and “evaluator size” in parentheses: 0.008 g (1.65), 0.02 g (2.36), 0.04 g (2.44), 0.07 g (2.83), 0.16 g (3.22), 0.4 g (3.61), 0.6 g (3.84), 1.0 g (4.08), 2.0 g (4.31), 4.0 g (4.56), 6.0 g (4.74), 8.0 g (4.93), 10.0 g (5.07), 15.0 g (5. 81), 26.0 g (5.46), 60.0 g (5.88), 100 g (6.10), 180.0 g (6.45), 300.0 g (6.65). Microfilaments were applied to the plantar surface of both hind paws (Figure 1c). A microfilament was placed perpendicularly to the skin with slowly increasing force until it bended. After the fiber bends, further advance creates more bend, but not more force of the application. The following protocol was used: 1.Adaptation of paw sensitivity to the mesh floor: We used the adaptation chamber () with a metal mesh floor (mesh size 0.5 cm) identical to that in the test chamber. The day before the test, rats were transferred from their home cage to the adaptation chamber for 1.5 h.2.Adaptation to the experimental environment: We used the test chamber a transparent plexiglass (30 cm × 20 cm) with a metal mesh floor (0.5 cm square size). The rat was placed in the test chamber for 15 min for adaptation.3.Test: The fiber was gently pushed against the surface of the skin from below. A positive response is a flinch of the leg indicating that it has clearly perceived the stimulus. Gentle pressure was applied to the fiber until the fiber was not bent at right angles. The interval between presentations was 5–10 s. The right and left paws were tested in an alternating order. The lower limit of the testing range was a 4.0 g (4.56) filament. The test started at 4 g, and filaments were used in the following sequence: 4, 6, 8, 10, 15, 26, 60, 100, 180, 300. The highest filament was 300.0 g (6.65), and some male subjects did not respond to it (Section 3.3). After the end of the test, the testing cage was cleaned with 50% ethanol solution to remove olfactory cues.4.The response was noted when the rat responded to 3 out of 5 presentations.5.In the left paw, we measured direct sensitivity as the first filament to which the rat responded (3 out of 5 presentations, as mentioned above).6.In the left paw, we computed the 50% withdrawal threshold as an accurate measure reproducible across laboratories [46,47,48]. The basis for estimating the 50% threshold is the binominal pattern of positive and negative responses to different stimulations. Accordingly, we used the range of filaments starting with a negative response (O) stimulus, and the next elicited a positive response (X). There was a certain pattern of negative and positive responses, e.g., OOOXOXXO, OOOOXOXXX. Individual patterns have a corresponding constant, k, provided in a reference table by Dixon [49]. Chaplan et al., in 1994, adapted the statistical insights of Dixon to an equation for the calculation of 50% withdrawal thresholds [48]:
$$T_{50\%}=10^{X_{f}+k\delta }/10000$$
where *X_f_* is the value of the final von Frey filament used (in log units); *k* is the tabular value given in Appendix 1 [48] (for the pattern of positive/negative responses); δ is the mean difference between stimuli in log units = 0.17 for rats [47].

#### 2.2.4. Active Avoidance Test

This test was performed in the standard two-way shuttle box, consisting of two compartments (30 × 26 × 30 cm) separated by an opaque wall with a passage in the middle. A grid floor in the shuttle box was made of 3 mm stainless steel bars spaced 1 cm apart, with a removable tray below the grid floor. A scrambled electric foot-shock (0.5 mA, 5 Hz) was used as an unconditioned stimulus. A pure tone (70 dB for 4.5 s) was used as a conditioned stimulus. Unconditioned and conditioned stimuli delivered automatically using customary computer software. An experimenter was in the next room and controlled the experiment using a computer system and a video camera (JVS GR-DVP7, China) located 180 cm above the shuttle box.

Adaptation: Rats were transferred to the experimental room and kept in cages for habituation (minimum 10 min). Each individual rat was placed in the shuttle box for 3 min to adapt to the environment. Ultrasonic vocalization was recorded during this adaptation period.

Testing: Rats were trained to avoid foot-shock by moving to the adjacent “safe” compartment after the presentation of a conditioned stimulus (4.5 s tone). A foot-shock was delivered in any compartment, and the opposite compartment was always “safe”. Electric foot-shock was delivered immediately after the offset of sound presentation and lasted until the rat moved to the opposite “safe” compartment. The reaction time was recorded automatically. If a rat did not respond during 40 s, electric stimulation was turned off automatically. Each rat performed 50 trials. The interval between trials was chosen randomly between 20 and 50 s. After the end of the training session, the camera was cleaned with a 50% ethanol solution to remove olfactory cues.

Rats demonstrated three main strategies: (1) avoidance when the learned reaction time was <4.5 s, i.e., the rat moved to the “safe” chamber before electric stimulation; (2) escape when the reaction time was >4.5 s (the rat moved to the “safe” chamber during electric stimulation); and (3) freezing when the rat stayed immobile and endured electrical current.

Ultrasonic vocalization was recorded during the entire test session.

### 2.3. Ultrasound Recording and Analysis

The Sonotrack system (Metris, Hoofddorp, The Netherlands) was used for the non-invasive recording and analysis of ultrasonic vocalizations. The system consisted of a Sonotrack Control Unit, ultrasound microphone, and the Sonotrack software v.2.6.2.30. Background noise was greatly reduced by software settings and the small opening angle of the applied microphones. With the Sonotrack system, we recorded the low-level ultrasonic vocalizations, performed full-spectrum analysis (15–100/125 kHz), and semi-automatically detected instant calls.

### 2.4. Statistical Analysis

Non-parametric tests were used: Mann-Whitney U test for independent variables and Wilcoxon matched pairs test for dependent variables. The parametric t-test with the Bonferroni adjustment for error rates was used in 50 tests. K-means cluster analysis verified by ANOVA, repeated measures ANOVA with Duncan post-hoc test, and Chi-square test were used where appropriate.

## 3. Results

### 3.1. Sucrose Preference Test 

Eleven females and 24 males were used in this experiment (Group 1, females and males, were littermates, the progeny of 6 mothers). The body weight in males was 378 g on average (min 331, max 423), and in females, it was 214 g on average (min 204, max 227). Both sexes preferred sucrose over water, and sucrose preference in female and male rats was the same (on the first day, −65 ± 34 and 59 ± 36; on the second day, −69 ± 25 and 87 ± 11, correspondingly). However, sucrose preference in males on the second day was higher than on the first day (*p* = 0.017, Wilcoxon Matched Pairs Test), but it was not in females (*p* = 0.50, Wilcoxon Matched Pairs Test, Figure 2a). The volume of 2% sucrose solution per body weight consumed on the first day was higher than on the second day in both sexes (both with *p* < 0.001, Wilcoxon Matched Pairs Test, Figure 2b). 

Next, we computed daily changes in body weight as a difference in body weight between two successive days. Females differed from males by losing their body weight on the first day of the sucrose preference test (*p* = 0.0018, Mann-Whitney U test, Figure 2c). Females on the second day of the experiment showed significantly less changes in body weight than those on the first day (*p* = 0.029, Wilcoxon Matched Pairs Test, Figure 2c). 

A high variability of daily changes of body weight in females and in males suggested strong individual differences in body weight during 2 days of sucrose consumption (perhaps due to different metabolic profiles). In order to classify the results, we performed a cluster analysis of daily weight gain measures using K-means clustering (Figure 2d). Two statistically significant clusters were defined in both sexes. 

Male rats represented two sub-groups with the opposite dynamics of body weight (marked by “++” in Figure 2d; ANOVA for day 1: F_1;22_ = 24.4, *p* = 6 × 10^−5^; for day 2: F_1;22_ = 39.1, *p* = 3 × 10^−6^). Males in cluster 1 (12 rats) on the first day showed weight gain that returned to the zero level on the second day. Males in cluster 2 (12 rats) showed the opposite weight dynamics (close to zero changes in body weight on the first day and weight gain on the second day).

Female rats were also classified in two clusters (marked by “++” in Figure 2d; ANOVA for day 1: F_1;9_ = 21.3, *p* = 0.0012; for day 2: F_1;9_ = 0.17, *p* = 0.69). Females in cluster 1 (7 rats) showed ~2 g lost weight on the first day and zero weight gain on the second day. Females in cluster 2 (4 rats) showed ~7 g lost weight on the first day and zero weight gain on the second day.

In general, adult male and female rats showed the same preference for sucrose, around 65–87%. Males, but not females, showed an increased sucrose preference on the second day (87%) compared to that on the first day (59%). Females, but not males, showed significant weight loss on the first day (3.6 g per rat on average). Considering the lack of sex-related differences in sucrose preference, a rapid decrease in body weight in females on the first day of sucrose exposure might be accounted for by an acute reaction to stress or by sex differences in metabolic processes.

### 3.2. A Free Left/Right Choice Task (T-Maze)

This test was performed on 11 females and 24 males (Group 1). The passive strategy was found in 3 males (12.5% of males), but it was not found in females (Figure 3a). The number of females with left, right, and no side preferences were 3, 4, and 4, correspondingly, and this did not differ from the number of males (4, 7, 10, χ^2^ (2, N = 32) = 0.45, *p* = 0.79). The laterality index in females and males did not differ (0.00 and 0.21 on average, correspondingly, *p* > 0.1, Mann-Whitney U test). This suggests no sex differences in behavioral lateralization.

### 3.3. Von Frey Test 

This test was conducted on 11 females and 24 males (Group 1). All females were in the diestrus stage of the estrus cycle (confirmed by vaginal cytology smears). Each rat was placed individually in the test cage for 15 min for adaptation to the experimental environment. The test session started immediately after the adaptation period and lasted from 3 min 23 s to 7 min 41 s (mean 4 min 16 s). Ultrasound vocalization was recorded during adaptation and test sessions. 

The lowest filaments to which we noticed responses were 8 and 10 g (4.93 and 5.07). The highest filament was the maximal available 300 g (6.65), to which 3 males did not respond. The median value of the “target force” was the same in males and females, 60 g. Left and right paws showed almost equal sensitivity to von Frey filaments. The 50% withdrawal threshold in females was in the lower quartile of threshold values in males (Figure 3b). 

The distribution of the withdrawal thresholds was evaluated, taking into consideration the outcomes of a free choice test (Figure 4a,b). Side preferences in a free choice test did not influence the withdrawal thresholds (all *p*”s in the Kruskal−Wallis ANOVA were >0.05), suggesting that left/right preference did not interfere with sensitivity to mechanical stimulation.

We noticed that the 50% withdrawal threshold exceeded 0.01 in 1 female (=0.014, 10% of all females) and in 11 males (46% of all males). The threshold value of 0.01 was used as a criterion to define subgroups with high (<0.01) and low sensitivity (>0.01). The difference between males and females comprising these subgroups was significant (Figure 4c, χ^2^ (1, N = 35) = 4.52, *p* = 0.033). Therefore, a low sensitivity to mechanical stimulation in males was found more often than in females.

### 3.4. Active Avoidance Test 

The test was performed on 17 females and 31 male subjects (Groups 1 and 2). All females were in the diestrus stage of the estrus cycle (confirmed by vaginal cytology smears). Rats were trained to avoid electrical stimulation (unconditioned stimulus) in a two-way shuttle box by moving to the adjacent “safe” compartment after the presentation of 4.5 s tone (conditioned stimulus, Section 2.2.4). This test was successfully performed by all females (100% of females) and 23 males (74% of males). The remaining 8 males (26% of males) demonstrated a freezing reaction (stayed immobile and endured electrical current) and were not able to learn associations between the tone and foot-shock. When freezing in males repeated for 15 trials, training was stopped. As long as freezing was found in a considerable number of subjects (26% of males or 16.7% of the total number of animals), we defined freezing as a behavioral strategy in an active avoidance task.

Test performance was analyzed in rats, which were able to learn (i.e., 17 females and 23 males). We measured the reaction time between the onset of a tone stimulus and the crossing to the next compartment (Figure 1d and Figure 5a). Reaction times of all 50 trials were used for classifying rats based on their performance with cluster analysis. Exploratory joining tree-clustering and K-means clustering demonstrated the presence of two clusters corresponding to the good and bad learners (Figure 5b,c). In males, 19 subjects (61% of males) comprised the group of good learners, and 4 (13% of males) were bad learners. In females, 13 subjects (76% of females) were good learners, and 4 (24% of females) were bad learners. Repeated measure ANOVA demonstrated a highly significant effect of the group factor (good/bad learners: F_1;36_ = 85.3, *p* = 4.9 × 10^−10^) and the effect of the sex factor (male/female: F_1;36_ = 64.4, *p* = 1.5 × 10^−9^) on the reaction time. The significant interaction of group*sex*trial (F_49;1764_ = 3.7, *p* = 1.4 × 10^−15^) suggested that male and females differed in learning profiles as shown in Figure 5b,c.

The Duncan post-hoc test revealed no significant difference between male and female good learners (all *p*”s > 0.05). Among females, bad and good learners showed different reaction times for the second, fifth, seventh, and ninth trials (*p*”s < 0.05, Duncan post-hoc test, Figure 5b,c). In males, differences between bad and good learners were found in the majority of trials. Moreover, the learning profile in female bad learners was similar to that in male good learners, with significant differences only for the first and ninth trials (*p* < 0.05, Duncan post-hoc test). 

Next, we analyzed reaction times, considering that avoidance reaction or successful learning (the rat avoided foot-shock) took less than 4.5 s and escape reaction took longer than 4.5 s (i.e., the rat passed to the “safe” compartment during electric stimulation). Differences from 4.5 s (avoidances) were estimated using a t-test with Bonferroni adjustment for error rates in 50 tests (*p* < 0.001 was set as significance level, “*” in Figure 5b,c). Reaction times, which did not differ from 4.5 s (“+” in Figure 5b,c), represented fast escape reactions or insufficiently learned avoidance reactions. Male good learners displayed insufficiently learned avoidance reactions between the 9th and 25th trials and showed significant avoidance reactions from the 26th trial to the end of the test. Female good learners displayed insufficiently learned avoidances between the 2nd and 11th trials and significantly defined avoidances from the 20th trial to the end of the test. Therefore, female good learners started learning earlier than males (2nd vs. 9th trial) and showed stable learned avoidances earlier than male good learners (20th vs. 26th trial). To generalize, female rats had better learning skills than males, and associative fear-conditioning in females was faster than in males.

### 3.5. Ultrasonic Vocalization

#### 3.5.1. Von Frey Test

Ultrasonic vocalization was recorded during the 15 min adaptation period and during the test session. Females did not produce ultrasonic calls either during the adaptation or during the test. In males, 2 out of 24 subjects emitted aversive ultrasonic calls >22 kHz during test sessions (Figure 3a) but not during the adaptation period. These rats started vocalizing in the second and fourth minutes of the test and continued until the end of the testing. The minimal frequency of aversive calls was on average 23.2 kHz; the mean frequency was 24.8 kHz (min 16.7 kHz and max 34.8 kHz).

#### 3.5.2. Active Avoidance Test

Ultrasonic vocalization was examined in 21 males and 9 females during a 3-min period of adaptation and during the entire test. During the 3-min adaptation, neither males nor females emitted aversive ultrasonic calls. During active avoidance learning, 11 males produced aversive ultrasonic calls (52.4% of the number of males), and 5 females produced them (55.6% of the number of females). The mean frequency of these calls in males and females did not differ and varied around 24–25 kHz (Figure 6b), but the sum length of aversive vocalization during test session in males was significantly higher than in females (Figure 6a). 

In general, around half of the males and half of the females emitted 24–25 kHz aversive calls during associative fear-conditioning, but ultrasonic vocalization in females took a shorter period of time than in males.

## 4. Discussion

In this study, we examined sex differences in littermates using positively and negatively motivated behavioral tasks in adulthood, i.e., between 5 and 8 months of age. Rats were derived from Wistar Albino Glaxo Rats from Rijswijk as a non-epileptic substrain, abbreviated NEW. This is a minor rat substrain selected from WAG/Rij. After the end of the behavioral experiments, Group 2 underwent electroencephalographic examination in free behavior, (Appendix A), and their non-epileptic phenotype was confirmed. For the reasons that (1) Group 2 contained parents of Group 1 and (2) the absence of epilepsy in an inherited disorder, NEW rats used in this study were considered as non-epileptic. Our team selected NEW rats as a non-epileptic control to WAG/Rij rats, and the parent strain was derived from Wistar albino stock [40,50]. Considering the common genetic background, the current behavioral findings in NEW rats could be applicable to the other rats of Wistar albino stock.

A battery of behavioral tests started with the positively motivated 2% sucrose preference test for two days. Here, we found the same preference for sucrose solution over water in males and females: on the first day (65 vs. 59%) and on the second day (69 vs. 87 correspondingly, Figure 2a). The volume of consumed 2% sucrose solution per body weight was also the same in females and males. These findings did not support the hypothesis that females had a stronger sucrose preference than males [6]. This disagreement might relate to methodological differences in test conduction. We did not deprive our rats from water and conducted this test for two days, 24 + 24 h. Perhaps female rats were more susceptible than males to overeat palatable/sweet food (e.g., [7]) rather than drink sweet solution. Our main conclusion is that the NEW rat substrain showed a relatively high preference for 2% sucrose, 59–67%. This might be interpreted as a hedonic behavioral response. 

It was surprising that females, but not males, lost weight on the first day of the sucrose preference test (Figure 2c, Appendix A). The most remarkable results were obtained in cluster 2 (4 females), who showed ~7 g weight loss on the first day. It is essential that all females showed almost zero weight gain on the second day, suggesting that sucrose consumption resulted in metabolic changes, which were fast. In total, sex differences in body weight in females might be accounted for by the emotional reaction to a novel subject (such as a new bottle with a palatable drink) and for fast processes of metabolic homeostasis. It seems likely that females lost their weight during the first test day because of increased locomotion (or behavioral excitation), reflecting curiosity. It is known that female rodents are characterized by a higher curiosity and higher locomotor activity in a novel environment than males [51,52,53]. A temporal increase in locomotion might cause a decrease in female body weight. The analysis of sex differences in locomotor activity in the sucrose preference test might give an answer to this question. Another longitudinal and high-throughput behavioral examination of sucrose preference was obtained using the TSE IntelliCage [54].

In a free-choice paradigm (T-maze, neutral situation), some males demonstrated a passive strategy (three subjects or 12.5% of males/8.5% of all subjects) that was not found in females. Left, right, and no side preferences were the same in males and females, around 1:1:1. The analysis of the lateralization index in the T-maze demonstrated no sex differences in behavioral lateralization. This corresponds well to results obtained in male Sprague-Dawley rats tested in the T-maze [55]; each rat preferred running either to the left or right side, but the difference between the total trials of running to left or right was not significant. However, as early as in 1991, behavioral lateralization in female and male Sprague-Dawley rats in the T-maze was studied by Josefina Alonso et al. [56]. They showed that the *«percentage of animals with right-preference (72%) was greater than the percentage of animals with left-preference (11%)».* No sex differences were found: *«most lateralized males (82%) and females (87%) had right-side preference»*. The above cited study [56] was performed in the electrified glass T-maze, where rats received a scrambled foot-shock at the start arm and were forced to escape by entering either the left or right arm. It was an aversive behavioral situation. In our study, rats in the T-maze were free to move and explore left/right arms or stay in the start arm. This suggests the absence of side preference in neutral behavioral situations.

In the current paper, we considered behavioral lateralization assessed in the T-maze (i.e., left, right, and no side preferences) for the analysis of mechanical sensitivity to von Frey hair stimulation. We initially hypothesized that lateralization would affect individual mechanical sensitivity, but we had to reject this hypothesis, because the results of the T-maze performance did not interfere with the sensitivity to mechanical stimulation (Figure 4a,b).

For the assessment of sensitivity to mechanical stimulation in the von Frey test, we computed the 50% withdrawal threshold in order to reach result comparability with other laboratories. We found a strong individual variability of this threshold; however, threshold values in females (during diestrus) were in the lower quartile of threshold values in males. The boundary value was determined as 0.01. More sensitive rats showed a withdrawal threshold below this value (90% of females and 54% of males). Less sensitive rats were above this value (10% of females and 46% of males). We concluded that the tactile threshold in females (during diestrus) was lower than in males. It is still questionable whether tactile sensitivity in females is influenced by hormonal changes during the estral cycle. 

Most often, the von Frey test in rats is performed to assess nociceptive processes in models of pain (see references in [46,48]). It is striking that females appeared to be more sensitive to mechanical stimulus, but they did not emit aversive calls during the von Frey test. Only 2 males (8% of males) emitted long aversive ultrasonic calls with the mean frequency of 24.8 kHz. Stefan Brudzynski suggested that the emission of rat 22 kHz calls represents the evolutionary vocal homolog of human crying [57]. It seems that 100% of females and 92% of males experienced no discomfort or pain in the von Frey test. At least, all females and the majority of males did not express vocally a putative aversive emotional state during von Frey test. Numerous human studies (reviewed by Adrian Maurer et al. [37]) demonstrated sex-specific differences in pain sensitivity and pain threshold. Sex-related differences in pain perception could be explained by complex biologic mechanisms, such as intranuclear mechanisms of action for steroid hormones, interactions on the level of hormone receptors and nociceptive receptors, proinflammatory mechanisms, and others [13,34,35,36].

The existing literature provides inconsistent results regarding the effects of sex in pain-related fear conditioning paradigms. Testosterone and its metabolites are known to link to reduced fear and anxiety in male rodents [19,58]. However, there are contradictory findings on the testosterone-mediated effects on anxiety-like behavior (see refs in [59]). As is known, females have a lower level of fear and anxiety than males (see references in [20]), and this might also affect fear conditioning. Another important concern is sex differences in emotional self-regulation. Here, we analyzed ultrasonic vocalization in aversive situations in order to characterize the emotional state of females and males. In our experiments, the mean frequency of aversive calls was 24–25 kHz, which is higher than 22 kHz ultrasonic calls, which are used as indicators of anxiety/distress (see references in [27]). The frequency of 22 kHz aversive calls is known to vary from 18 to 32 kHz [27,33,60]. Here, we measured the frequency properties of thousands of ultrasonic calls with Sonotrack Metris software and presented the average results. There might be several frequency specific classes of aversive calls, and differences might also be found between subjects with low and high localization levels.

Our data indicated that males more frequently emitted 24–25 kHz aversive calls during fear conditioning than females (in diestrus). Therefore, males more often expressed their aversive emotional state vocally. Similarly, Rebeca Machado Figueiredo et al. [33] noted that females during isolation restraint emitted far fewer 22 kHz aversive ultrasonic calls than males, suggesting that females found this experience less aversive or were less stressed. In addition, females produced three times more 22 kHz calls in late diestrus than during other stages of the estral cycle [33]. It is striking that the number of aversive ultrasonic calls in females in late diestrus and in males were the same [33], suggesting that emotional reaction to stress in females (late diestrus, but not in other stages) and in males was similar.

Freezing in rodents is a specific behavioral response to an unpredictable threat, such as a foot-shock in a laboratory or in a natural situation with a high risk of predation. Here in the fear-avoidance task, we found that around 26% of males, but no females, experienced frizzing and were not able to learn. This type of reaction might be classified as helplessness-like behavior. It fits well with the literature, demonstrating that female rats tend to freeze less than males (see references in [26]). 

In the current experiments, we defined two learning profiles in males and females—good and bad learners. In females, 76% were good learners, and 24% were bad learners. In males, good learners comprised 61% of males, bad learners comprised 13% of males, and 26% were not able to learn. We defined fast escape reactions, which took around 4.5 s (duration of a conditioned tone stimulus) and might be interpreted as insufficiently learned avoidance response. Based on the statistical analysis of fast escape reactions and avoidances, we concluded that the learning profile in female bad learners was similar to that in male good learners. Similarly, Tara Chowdhury et al. [23] indicated that initial instrumental avoidance learning in females was significantly faster than in males. It is remarkable that these authors defined two behavioral strategies in females [23]: *«about half of females successfully avoided the shock (“avoiders”), the other half consistently waited for shock to begin before immediately performing the response to turn it off (“escapers”)».* In their experiments, males demonstrated *«a near-uniform elevation in freezing to an aversive conditioned stimulus».* Here, we used a standard associative fear-conditioning paradigm, and rats that successfully performed the test (100% females and 61% of males) showed both escape reactions and avoidances. In these rats, we evaluated performance progress in 50 trials, from fast escape to avoidance. 

The shortcoming of our study is different numbers of females and males, 17 and 31 correspondingly. The full behavioral test battery was conducted on rats of Group 1, which were siblings from five litters born in March–April 2022. The number of pups varied from 5 to 9 per litter, but the number of males exceeded the number of females. The strong point is that Group 1 rats were sisters and brothers and they were born during the same short period of time; therefore, genetic and epigenetic variations within Group 1 were minimal. Another limitation concerns the sucrose preference test. We measured sucrose preferences in 2–3 rats per cage and could not assess individual preferences. This was a positively motivated task, in which social isolation for 2 days was not acceptable. Nevertheless, the results of the entire test battery in each subject describe a complex behavioral phenotype. This behavioral phenotype may be interpreted as the individual neurocognitive profile. Creating individual neurocognitive profiles might be one of the future directions to better evaluate in vivo rat models of neuropsychiatric disorders.

## 5. Conclusions

Our study was performed on a NEW rat substrain, i.e., non-epileptic WAG/Rij rats of ages between 5.8 and 7.6 months. NEW rats showed relatively high preference for 2% sucrose around 59–67%. This might be interpreted as a hedonic behavioral response, in contrast to anhedonia defined in epileptic WAG/Rij rats [61]. No sex differences were found in the sucrose preference test; however, females lost weight on the first day of this test perhaps due to their reaction to acute stress or processes of metabolic homeostasis.

In a neutral behavioral situation, the T-maze, rats were free to move and explore left/right arms or to stay in the start arm. Males and females demonstrated similar behavioral lateralization, around 1:1:1 for left, right, and no side preferences.

In two aversive tests (von Frey test and associative fear conditioning), females were in diestrus as defined with vaginal smears. In order to characterize the emotional state in these aversive tests, we analyzed ultrasonic vocalization and, in particular, 24–25 kHz aversive calls. Males more frequently emitted 24–25 kHz aversive calls than females, suggesting that males more readily vocally expressed their aversive emotional state then females.

We measured tactile sensitivity in a von Frey test by means of the 50% withdrawal threshold in order to reach result comparability with other laboratories. More sensitive rats showed a withdrawal threshold below 0.01 (90% of females and 54% of males), and less sensitive rats showed one above 0.01 (10% of females and 46% of males). Therefore, the tactile threshold in females (during diestrus) was lower than in males.

In the associative fear-avoidance task, we found that around 26% of males, but no females, experienced frizzing and were not able to learn. This type of reaction might be classified as helplessness-like behavior. We defined two learning profiles in males and females—good and bad learners. The ratio good:bad learners in females was 76:24% and males 61:13% (26% of males were not able to learn). Based on the statistical analysis of fast escape reactions and avoidances, we concluded that the learning profile in female bad learners was similar to that in male good learners. In general, female rats had better learning skills than males, and associative fear-conditioning in females was faster than in males.

Our results might contribute to behavioral neuroscience in rodents and might be helpful for the in vivo modeling of neuropsychiatric disorders. A complex behavioral phenotype based on the results of the entire test battery in each subject can characterize the individual neurocognitive profile. Creating individual neurocognitive profiles might be one of the future directions to better evaluate the in vivo rat models of neuropsychiatric disorders.

## Figures and Tables

**Figure 1 life-13-00547-f001:**
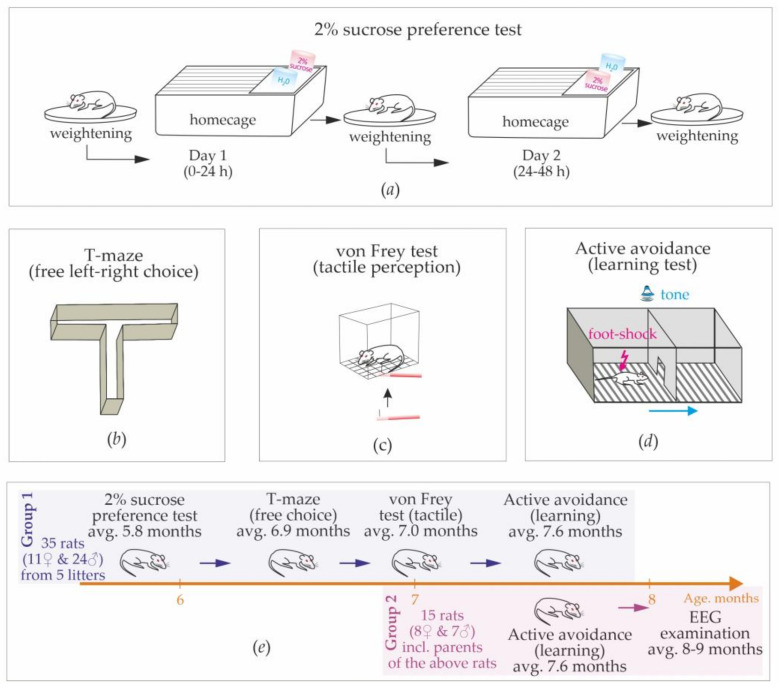
The sequence of behavioral tests (**a**–**d**) and timeline of experiments (**e**).

**Figure 2 life-13-00547-f002:**
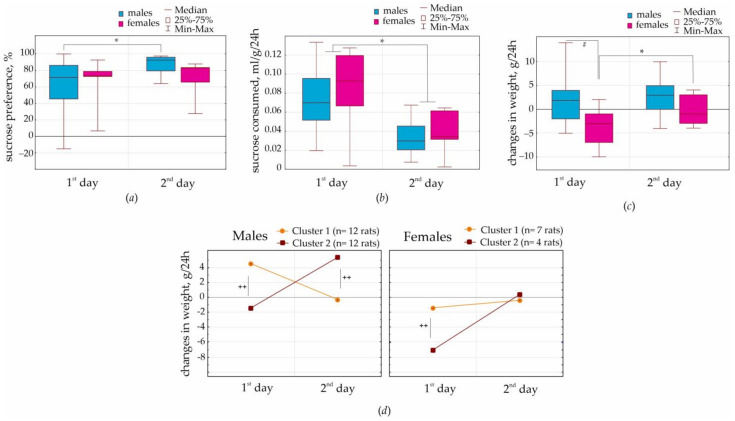
Results of the 2% sucrose preference test performed for 2 days (24 + 24 h) in females and males. (**a**) Sucrose preference in %; significant differences are shown by * *p* < 0.05 Wilcoxon Matched Pairs Test (between two subsequent days). (**b**) Sucrose consumption computed per 1 g of body weight during 24 h interval; * *p* < 0.001 Wilcoxon Matched Pairs Test (between two subsequent days). (**c**) Daily changes in body weight in g per day 24 h; significant differences between males and females are shown by # *p* < 0.05 Mann-Whitney U test. (**d**) Results of the cluster analysis of daily changes in body weight, indicating the presence of two significantly different groups with different dynamics of daily changes in body weight ^++^ (ANOVA, *p* < 0.05).

**Figure 3 life-13-00547-f003:**
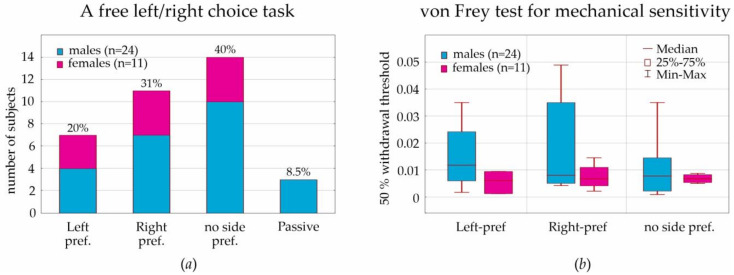
Results of two behavioral tests in males and females: a free choice task based on 5 trial scores in the T–maze (**a**) and the von Frey test 50% withdrawal threshold for mechanical stimuli applied to the left hind limb (**b**).

**Figure 4 life-13-00547-f004:**
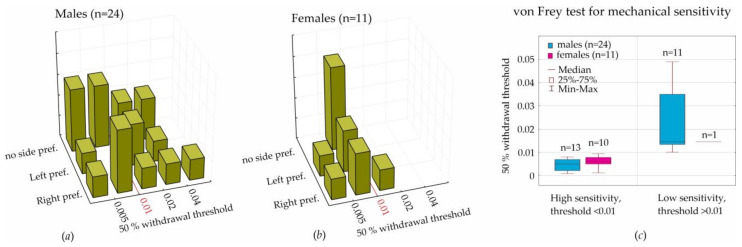
Statistical results of the free choice test (three strategies of choice: left preference, right preference, or no side preference) and von Frey test (the 50% withdrawal threshold). Distribution of the 50% withdrawal threshold as determined in subjects with left, right, or no side preference: males (**a**) and females (**b**). The value of the 50% withdrawal threshold is shown in subjects with high and low sensitivity (**c**).

**Figure 5 life-13-00547-f005:**
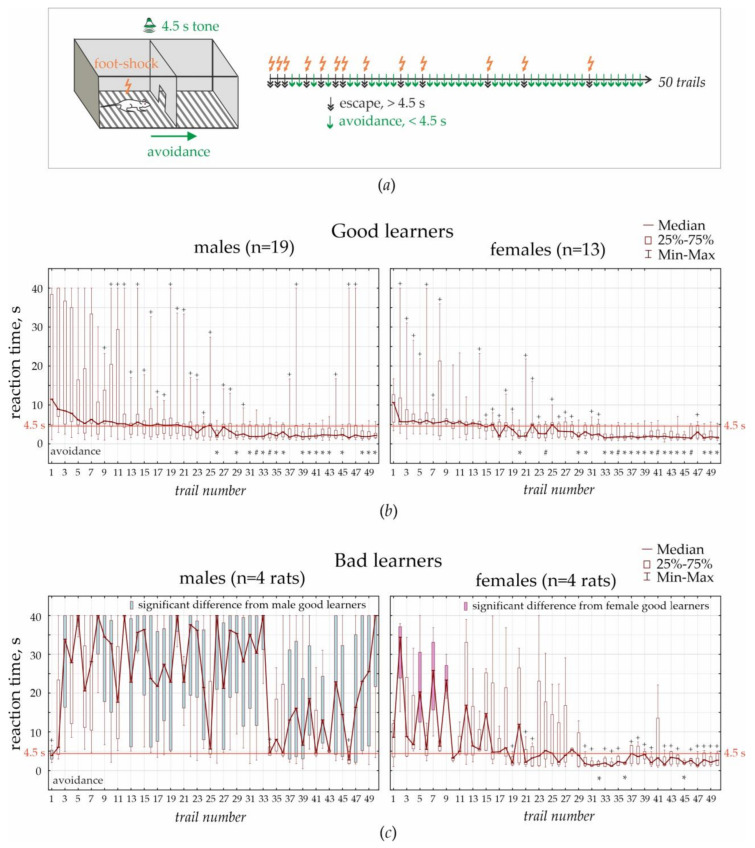
Active avoidance test: (**a**) The scheme of a two-way shuttle box in which the rat was presented with the conditioned (tone for 4.5 s) and unconditioned (scrambled electric foot-shock) stimuli. The rat trained to avoid the foot-shock by moving to the adjacent “safe” compartment on the tone presentation. The test consisted of 50 trials shown by short arrow-markers on the long horizontal arrow. Avoidance (green arrow-markers) was the learned reaction to avoid electric foot-shock during tone presentation (reaction time < 4.5 s); escape meant movement to the “safe” compartment during electric stimulation after ceasing the tone. (**b**,**c**) Results of the active avoidance task in females and males. Rats were classified as good and bad learners using K-means cluster analysis. In reaction times, significant differences of 4.5 s (avoidances) are indicated by “*” (*p* < 0.001 Bonferroni adjustment for error rats in 50 tests); #—*p* < 0.01. Non-significant differences of 4.5 s are indicated by “+” pointing to unstable responses or fast escape reactions. Filled boxes demonstrate significant differences between good and bad learners in each sex independently (Duncan post-hoc test, *p* < 0.05).

**Figure 6 life-13-00547-f006:**
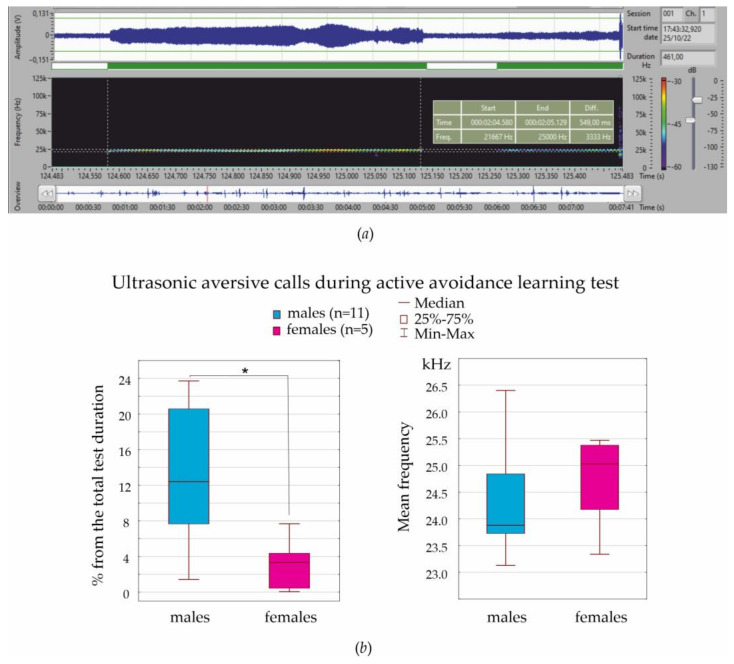
Ultrasonic vocalization emitted in aversive behavioral situations: (**a**) An example of a 21.6−25 kHz ultrasonic aversive call emitted by a male during the von Frey test. The signal was recorded and processed with Sonotrack software. The upper plot demonstrates the amplitude spectrum that was used for the semi−automatic detection of calls. (**b**) Characteristics of ultrasonic aversive calls in males and females during the active avoidance test. * significant differences between males and females (*p* < 0.05, Mann-Whitney U test).

## Data Availability

All experimental data obtained in the current study are shown in figures and tables. Primary datasets are available from the corresponding author upon reasonable request.

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
