# Peer review of "Sex Differences in Behavior and Learning Abilities in Adult Rats"

_life, 2023, doi:10.3390/life13020547_

Round 1

Reviewer 1 Report

In the manuscript by Pupikina and Sitnikova authors report the results of their studies on behaviour and learning differences in male and female adult rats. Despite a growing body of evidence supporting the necessity to include groups of both sexes in neurocognitive studies with a separate analysis of each group, this clearly important area is not sufficiently researched. Knowledge about sex-driven baseline difference in behaviour and learning is particularly important for the drug development research where male-only studies are still prevalent.

The authors used a combination of tests to assess and compare behaviour and learning abilities of male and female rats. That included a sucrose preference test, T-maze (a free-choice paradigm) and tactile sensitivity examination by Frey fibres. They have shown that consumption of 2% sucrose during the first day of testing led to the weight gain in females’ group but not in males. Authors explained this by differences in metabolic homeostasis. They also provided evidence that males had lower tactile sensitivity than females and performed differently in the associative fear-avoidance task.

Overall, this is a well-designed high-quality research, controls used are adequate. Nevertheless, there are several minor issues that authors need to address:

Sucrose consumption and body weight vary significantly between individual animals. Therefore, authors have performed the cluster analysis of daily weight gain measures and revealed two groups in both sexes:

1.         It would be helpful to have in the Introduction more detailed discussion of data available in the literature on individual clustering of rodents participated in this common 2% sucrose preference test (positive motivation).

2.         In von Frey the tested animals also were divided in two groups for high and low sensitivity (13 and 11). In sucrose test the same males were clustered 12 and 12. Were there any correlations between the presence of individual animals in high/low sensitivity and sucrose preference groups?

3.         T−maze test revealed 8% of males were passive in a free choice task. Where there any overlapping animals with either of above groups?

Providing in Supplementary individual numbers for the subjects after clustering would be appropriate.

4.         Authors used a new rat substrain, non-epileptic WAG/Rij rats, which has been derived from epileptic strain. Animals’ non-epileptic phenotypes have been confirmed by EEG at the end point of experiments. These data should be included as Supplementary material.

Author Response

Dear Reviewer,

Thank you very much for your careful review and constructive recommendations. We highly appreciate the time and effort that you dedicated to evaluating our paper. Please find our point-to-point answers below.

Sucrose consumption and body weight vary significantly between individual animals. Therefore, authors have performed the cluster analysis of daily weight gain measures and revealed two groups in both sexes:

  1. It would be helpful to have in the Introduction more detailed discussion of data available in the literature on individual clustering of rodents participated in this common 2% sucrose preference test (positive motivation).

Response. We are very grateful for this idea. We inserted Table B1 in Supplementary B, where we reported individual data obtained in the sucrose preference test. We found a drastic mistake in the computations of body weight difference. When computing a daily difference, i.e., “a difference in body weight between two successive days”, we reversed Day 0 and 1 (same for Day 1 and 2), and therefore the difference was just the opposite to what initially reported. Clusters were exactly the same, but with mirrored scores. Thank you very much for the suggestion to show individual data, otherwise we would be faced with difficulties in the future. In the revised version we corrected results (text and Figure 2) and changed discussion.  

  1. In von Frey the tested animals also were divided in two groups for high and low sensitivity (13 and 11). In sucrose test the same males were clustered 12 and 12. Were there any correlations between the presence of individual animals in high/low sensitivity and sucrose preference groups?

Response. Thank you for this idea. Correlation between outcomes of different tests in one animal is an exciting issue. The problem is that sucrose preference was measured in 2-3 rats per cage, and we were not able to assess individual preferences. The outcomes of other tests showed no significant correlations (or the number of subjects was too low to reach the level of significance). “Nevertheless, the results of the entire test battery in each subject describe a complex behavioral phenotype that can better characterize the individual neurocognitive profile.” This idea has been added to Discussion of future directions. “Creating individual neurocognitive profiles might be one of the future directions to better evaluate in vivo rat models of neuropsychiatric disorders.”

  1. T−maze test revealed 8% of males were passive in a free choice task. Where there any overlapping animals with either of above groups?

Response. These animals did not show any peculiar results in the other tests. There were 3 males (8.5% of the total number of rats) that were passive in the free choice task (now in Supplementary, Table B2). Two of them showed low 50% threshould and one – high 50% threshould. Two of them gained weight in the sucrose preference test, and one, in the opposite, lost their weight.

Providing in Supplementary individual numbers for the subjects after clustering would be appropriate.

Response. We added Supplementary B with Tables B1 and B2, where we reported individual results as well as results of cluster analysis.

  1. Authors used a new rat substrain, non-epileptic WAG/Rij rats, which has been derived from epileptic strain. Animals’ non-epileptic phenotypes have been confirmed by EEG at the end point of experiments. These data should be included as Supplementary material.

Response. We briefly introduced this minor substrain on page 3 and our motivation to use them (also in discussion, page 15). We also added Supplementary A with four Figures, in which we briefly explained the procedure of EEG recording and analysis in NEW rats (vs WAG/Rij rats).

Reviewer 2 Report

The aim of the paper is to study sex differences in behavior and learning abilities in adult rats.

The paper presents a high quality of processing in terms of methodology, contents and results.  As regards the introduction, it would be appropriate to increase the number of references and the length of the introduction, also introducing the background paragraph to provide a broader overview of the existing scientific literature also through a rapid revision of the same. In particular

in lines 45-49 there is an essentially short passage from the first studies on the subject of behaviors connected to sex, to the more recent ones. It would be appropriate to analyze this aspect more in depth, highlighting previous studies also through a summary table of these eureka moments.

From a methodological point of view,

the inclusion and exclusion criteria of the rats should be expressed more clearly, and in general the whole recruitment process should be represented, sep by step, with a flow diagram.

The results are well represented both graphically and from a descriptive point of view

The discussion is well structured, but I would suggest to increase the references because they are very few compared to the contents reported.

A specific paragraph should be added with the strengths and limitations of the study.

 The conclusions must also be better specified by writing a paragraph entitled conclusions, making known what future research developments may be.

Moderate editing of English language and style is required

Author Response

Dear Reviewer,

Thank you very much for your careful review and constructive recommendations. We highly appreciate the time and effort that you dedicated to evaluating our paper. Please find our point-to-point answers below.

in lines 45-49 there is an essentially short passage from the first studies on the subject of behaviors connected to sex, to the more recent ones. It would be appropriate to analyze this aspect more in depth, highlighting previous studies also through a summary table of these eureka moments.

Response. Sucrose preference tests have a long history of use in rodent studies. In the revised paper we mentioned that “…  sucrose preference test has been widely used in rodents in many variations [4-6]”. Than we focused on relevant recent studies. In addition, we added part (e) in Fig. 1 showing a summary of the tests used in the current report.

From a methodological point of view, the inclusion and exclusion criteria of the rats should be expressed more clearly, and in general the whole recruitment process should be represented, sep by step, with a flow diagram.

Response. There was no recruitment process. We cannot specify either inclusion or exclusion criteria, because we tested all rats. A battery of behavioral tests was done in Group 1 (35 rats) from 5 litters, including 11 females and 24 males. Two female rats from Group 1 were lost and were not used in active avoidance tests. Active avoidance learning was examined in Group 1 (33 rats: 9 females and 24 males) and in Group 2 (15 NEW rats of the same age: 7 males and 8 females). We corrected the number of rats and clarified this issue in Methods on page 3. A flow diagram has been added to Figure 1(e).

The discussion is well structured, but I would suggest to increase the references because they are very few compared to the contents reported.

Response. Thank you for this hint. We better discussed behavioral results and future directions, and added more references to related works. The number of references is now increased from 53 to 61.

A specific paragraph should be added with the strengths and limitations of the study.

Response. At the end of Discussion on page 16 we added a paragraph with limitations, strengths of this study and future directions. “The shortcoming of our study is different numbers of females and males, 17 and 31 correspondingly. The full behavioral test battery was done in rats of Group 1, which were siblings from 5 litters born in March-April 2022. The number of pups varied from 5 to 9 per litter, but the number of males exceeded the number of females. The strong point is that Group 1 rats were sisters and brothers and they were born during the same short period of time, therefore, genetic and epigenetic variations within Group 1 were minimal. Another limitation concerns the sucrose preference test. We measured sucrose preferences in 2-3 rats per cage and could not assess individual preferences. This was a positively motivated task, in which social isolation for 2 days was not acceptable. Nevertheless, the results of the entire test battery in each subject describe a complex behavioral phenotype. This behavioral phenotype may be interpreted as individual neurocognitive profile. Creating individual neurocognitive profiles might be one of the future directions to better evaluate in vivo rat models of neuropsychiatric disorders.”

The conclusions must also be better specified by writing a paragraph entitled conclusions, making known what future research developments may be.

Response. We summed up our main findings in Conclusions and added the paragraph, in which we specified future research developments. “Our results might contribute to behavioral neuroscience in rodents and might be helpful for in vivo modeling of neuropsychiatric disorders. a complex behavioral phenotype based on results of the entire test battery in each subject. This can characterize the individual neurocognitive profile. Creating individual neurocognitive profiles might be one of the future directions to better evaluate in vivo rat models of neuropsychiatric disorders.”

Moderate editing of English language and style is required

Response. The English has been revised throughout the text.

Reviewer 3 Report

Pupikina et al. investigated how cognitive function in rats differs between males and females and found that generally speaking, female rats are superior to males when neurocognitive research is conducted.

Overall, this research is already good enough for publication, but needs to be corrected in some minor points as follows.

L126: 'data not shown' -> The electrophysiological data/evidence should be presented.

L298: 'Figure 2. Results 2%' -> 'Figure 2. Results of 2%'

L585: This work was done using a NEW rat substrain, but this strain is not famous or common in use of the neurophysiological/behavioral research, to the best of my knowledge. The authors should mention that this is a minor strain in the world and explain the possibility that the current findings will be applicable to other rat strains in general based on some certain specific reasons.

Throughout the text, P values should be presented in a style of 'P ~ 0.03', or 'P = 4.0 × 10-3', not a style of 'P < 0.001', unless the statistics produced only P < 0.05.

Author Response

Dear Reviewer,

Thank you for your positive feedback. We highly appreciate the time and effort that you dedicated to evaluating our paper. Your comments concerned a better introduction of NEW substrain, electrophysiological data (in Supplementary) and reporting statistical significance. Please find our point-to-point answers below.

 L126: 'data not shown' -> The electrophysiological data/evidence should be presented.

Response. In the revised paper we have added Supplementary A with four figures, in which we briefly explain the procedure of EEG recording and analysis in NEW rats (vs WAG/Rij rats). In Table B1 (Supplementary) we report individual results of EEG analysis in Group 2.

L298: 'Figure 2. Results 2%' -> 'Figure 2. Results of 2%'

Response. Thank you. This error is corrected, as well as many other errors in style and grammar.

L585: This work was done using a NEW rat substrain, but this strain is not famous or common in use of the neurophysiological/behavioral research, to the best of my knowledge. The authors should mention that this is a minor strain in the world and explain the possibility that the current findings will be applicable to other rat strains in general based on some certain specific reasons.

Response. Thank you for this idea. We added the last paragraph to Introduction for explaining our rationale. “Our study was done in a non-epileptic rat substrain derived from Wistar Albino Glaxo Rats from Rijswijk (Non-Epileptic WAG/Rij abbreviated as "NEW"). The mother strain of NEW rats, WAG/Rij rats are well-accepted rat model of absence epilepsy showing spontaneous spike-wave discharges in their in their electroencephalogram, EEG [39,40]. Selection work and breeding of NEW substrain has been carried out in our institution (Institute of Higher Nervous Activity and Neurophysiology of RAS, Moscow) since the second decade of the 21st century, when we selected female and male WAG/Rij rats without seizures during the entire life [41]. NEW rats were selected as a non-epileptic control to WAG/Rij rats and had similar genetic background. Considering the common genetic background, the current findings in the NEW rats could be applicable to the other Wistar Albino rat strains.”

Throughout the text, P values should be presented in a style of 'P ~ 0.03', or 'P = 4.0 × 10-3', not a style of 'P < 0.001', unless the statistics produced only P < 0.05.

Response. Corrected.

Reviewer 4 Report

The authors raised an excellent point about the behavioral test, as male rats are used more than females. This is fantastic work and will be helpful for researchers in rodent behavior research. 

I have minor query:

1. The authors should include the details of the strain of rats used in their study. Sometimes the behavioral results are also varied due to strain differences. 

2. Please mention how many rats were kept in the same cage. Sometimes, the number of rats in one cage also affects the behavior. 

Author Response

Dear Reviewer,

We are deeply grateful for your decision on our manuscript. Thank you for your positive feedback. We corrected the MS in accordance with your remarks.

  1. The authors should include the details of the strain of rats used in their study. Sometimes the behavioral results are also varied due to strain differences. 

Response. Thank you for this suggestion. In the revised manuscript we added one paragraph, which briefly introduced the NEW substrain. “…NEW rats were selected as a non-epileptic control to WAG/Rij rats and had similar genetic background. Considering the common genetic background, the current findings in the NEW rats could be applicable to the other Wistar Albino rat strains.”

  1. Please mention how many rats were kept in the same cage. Sometimes, the number of rats in one cage also affects the behavior.

Response. Indeed, housing is an important factor influencing rat behavior. Here our rats were housed 3-4 per cage. This is now mentioned in Methods (page 3).

Round 2

Reviewer 3 Report

Thank you very much for all of the corrections. I do not have any further comments or concerns.